# Playing in ‘*The Backyard*’: Environmental Features and Conditions of a Natural Playspace Which Support Diverse Outdoor Play Activities among Younger Children

**DOI:** 10.3390/ijerph191912661

**Published:** 2022-10-03

**Authors:** Janet Loebach, Adina Cox

**Affiliations:** 1Department of Human Centered Design, Cornell University, Ithaca, NY 14853, USA; 2Department of Landscape Architecture, Iowa State University, Ames, IA 50011, USA

**Keywords:** outdoor play, nature play, natural playspaces, play environments, physical activity, risky play, loose parts, topography, behavior mapping

## Abstract

Outdoor play in nature-rich spaces has been associated with healthy development among young children. The diverse play opportunities afforded to children by natural playspaces can scaffold health benefits, appreciation of nature, and pro-environmental behaviors into adulthood. Environmental features and conditions of outdoor playspaces significantly influence the diversity and quality of play opportunities. Understanding how the physical environment can support high-quality play experiences can inform the design of stimulating, health-promoting playscapes for children. An observational behavior mapping framework was utilized to examine the environmental features of *The Backyard*, a large natural playscape, associated with play activities among young children. The *Tool for Observing Play Outdoors* was used to capture outdoor play types OPT), along with associated behavioral and environmental data, during seven days of field observation. While the playspace supported most OPTs, Physical and Exploratory play were most prevalent. Associations with activity intensity and risk play are also presented. Loose parts, particularly natural loose parts, were highly involved in most OPTs, but especially associated with Exploratory play. Ground topography showed some association with several OPTs and warrants further investigation. The environmental features of *The Backyard* supported an abundant and diverse range of outdoor play activities for young children and families.

## 1. Introduction

The last ten or fifteen years of research and practice have firmly established the benefits of time outdoors for children’s healthy development and wellbeing, particularly outdoor play in nature-rich spaces [1,2,3,4,5,6]. In the same time frame we have also seen increasing evidence that children in Western countries are spending less time outdoors [7,8,9,10], and many do not have access high-quality outdoor playspaces, or have the opportunity for regular contact and engagement with nature [3,11,12,13], an inequity which was highlighted during the COVID-19 pandemic [14,15,16,17,18].

Outdoor play has been associated with key developmental advancements, and a growing body of research confirms that the physical environment of a playspace significantly influences play value and quality. Understanding the environmental features and conditions of playspaces which can prompt and scaffold high-quality play experiences can help inform the (re)design of stimulating, health-promoting playscapes for children.

### 1.1. Support for Physical Activity and Literacy

Well-designed natural playspaces can support the development of physical and movement skills in children, including fitness, endurance, coordination and fine motor control [1,19,20]. A recent systematic review demonstrated that more naturalized playscapes in particular appear to increase both the variety and physical intensity of children’s movement activities, supporting both gross and fine motor development [5]. Wishart et al. [21], comparing affordances for active play for preschoolers in a traditional versus a naturalized playscape, found that the naturalized space offered a greater variety of opportunities to engage in physical activity and to develop a sense of balance. There is growing research support which ties increased physical activity or literacy to both the specific environmental features of a playspace such as larger open spaces, well-connected pathways, and harder surface areas [22,23,24,25] as well as the diversity of opportunities or prompts for active play available [26,27,28,29,30].

### 1.2. Supporting Opportunities for Risky Play

The importance of positive risk opportunities in children’s play environments and activities has been well-established by research over the last 10 to 15 years. By attempting to navigate challenging, but not hazardous, situations or conditions encountered in their environment, children in fact learn how to be and stay safe [31,32,33,34]. Risk which is experienced through the vehicle of play allows children to examine their own capabilities and limits under safer, lower consequence conditions, presenting less threat to their physical and emotional well-being [34,35,36,37]. Risky play has been associated with positive health and development outcomes including improved self-esteem, self-regulation and resilience, as well as increased motor skills, agility and environmental competence [34,36,38]. Risk experienced through play can also significantly increase children’s ability to assess and cope with risks that they will continue to encounter throughout their life [33,39].

Opportunities for risky play, however, have also been tied to available features in the built environment. Numerous studies have reported greater opportunities for and engagement in positive risk play behaviors in more naturalized play spaces [40,41]. The varied materials and landforms, such as craggy boulders and uneven and sloped surfaces, that tend to be present in natural playscapes also appear to encourage risky play activities [20,41,42,43]. Loose parts, including natural parts such as mulch, acorn, stumps and stones, have also been associated with engagement in risky play [5,32,34]. 

### 1.3. Environmental Features Which Support Richer Outdoor Play Activities

The type and quality of play can be significantly influenced by the physical environment of a playspace [20,24,41]. Outdoor environments tend to offer a more enriched environment for play, affording more varied and less structured play opportunities, which in turn can stimulate creativity, facilitate learning and support physical development [8,19,26,41,44]. Outdoors spaces which provide opportunities for engagement with nature and natural materials have been shown to foster more varied and complex play, as well as positively contribute to children’s physical, emotional and social development [20,21,26,41,45]. Environments which provide more diverse play affordances can also engage a broader spectrum of children and for longer periods of time [39]. 

### 1.4. Loose Parts

The quality and richness of play opportunities may be improved by the provision of loose parts. Simon Nicholson [46] (p. 6), in his Theory of Loose Parts, stated that “In any environment, both the degree of inventiveness and creativity, and the possibility of discovery, are directly proportional to the number and kind of variables in it”, and further explains that children find joy in playing, experimenting, discovering, and inventing with loose parts. Loose parts can provide higher levels of engagement in outdoor environments, and provide a wide variety of play opportunities that include social interaction, language use, risk taking and inclusivity [46]. Loose parts have been shown to contribute to opportunities for constructive play, as well as allowing children opportunities to construct their own spaces, which are often used for imaginative play episodes such as building ‘houses’ or ‘forts’ [47]. Loose parts can also provide opportunities for open-ended, non-scripted play and have been associated with creativity, imagination, problem-solving, physical activity, and risk taking behaviors and thus are an integral part of nature playspaces [1,32,48,49,50]. Loose parts which can be provided within outdoor environments can include manufactured elements, such as balls, buckets, shovels, and kitchen equipment or tools, as well as materials often found in nature, such as pinecones, rocks, sticks, mulch and stones [50]. Additionally, outdoor play environments have the ability to provide generous sand and water elements which can lead to novel and stimulating play opportunities [50].

### 1.5. Diverse Topography

Topography may also be an important environmental design consideration due to the variety of affordances that may be offered from variances in topography, such as climbing, running, rolling, etc. Fjørtoft [19] found that moving on rugged ground by climbing trees and rocks improved motor fitness in preschool children. A study comparing a naturalized playground to a traditional playground found that uneven and irregular natural surfaces, including variations in slope and heights, afforded more balancing opportunities than the traditional playground [21]. Open ground (flat, relatively smooth surface) affords opportunities for running, walking and games, while sloping terrains provide affordances for rolling, sliding, and clambering [51]. Playspaces which include ‘cliffs’ or edges of differing heights also has shown to encourage positive risky behaviors such as jumping from and climbing up [52].

### 1.6. High-Quality Playspaces for Young Children

Exposure to enriched, high-quality playspaces may be especially crucial in the early years, for children under the age of 7 or 8 years. It is becoming increasingly apparent that activities and experiences that affect the very youngest of children can have lifelong consequences into adulthood. The ages of 0–3 years are particularly important for early brain development [53]. A technical report from the American Academy of Pediatrics, recognizing that the effects of early childhood experiences are responsible for lifelong consequences for educational achievement, economic productivity, health and longevity, has proposed a new ecobiodevelopmental (EBD) framework that can guide a deeper understanding towards health outcomes [54]. This EBD framework includes the understanding that Ecology, in this case the social and physical environment of the child, works in conjunction with Biology and Health and Development towards developmental outcomes and life course trajectories [54]. Shonkoff and colleagues stated that “longitudinal studies that document the long-term consequences of childhood adversity indicate that alterations in a child’s ecology can have measurable effects on his or her developmental trajectory, with lifelong consequences for educational achievement, economic productivity, health status, and longevity” [55] (p. e234). A recent systematic review of studies considering the influence of naturalized playspaces on young children demonstrate that unstructured nature play has a positive impact on their physical activity and cognitive development, particularly as it affords more imaginative play [43]. This work emphasizes the value of determining the environmental features and conditions of playspaces which can encourage high-quality play among the youngest groups of children.

### 1.7. Value of Nature Play in a Changing Climate

Knowledge of natural processes and positive attitudes toward nature will be important in addressing the world’s climate and biodiversity crises. Children’s early positive experiences in nature have been associated with development of a sense of connection to nature [56,57,58,59]. Young adults who spent more time in nature were found to have spent more time outdoors as children [58,60]. The myriad benefits that adults experience when spending time spent in nature may therefore be lessened for those deprived of beneficial positive childhood nature experiences. Substantial evidence also links time spent in nature, particularly during childhood, to pro-environmental attitudes and behaviors and appreciation for the natural world [58,61]. The United Nations, in a report on the workshop for Biodiversity and Climate Change [62], recognizes that the willingness and ability for people to adapt to new cultural values and behaviors will be an important role in society’s ability to mitigate the climate and biodiversity crises that will affect the quality of life for humanity.

The naturalization of playspaces, whether in public spaces, schools or child-care facilities, may therefore be critical interventions in the face of a changing climate, where extreme weather and temperatures are making it more difficult for children to play outdoors for long periods or at certain times of year. Natural playspaces with significant tree canopy and plant materials, as well as natural ground surfaces, can provide shaded areas for continued play without excessive UV exposure, as well as cool the ambient temperature of the playspace to lower or limit heat stress experienced from hardscape materials such as metal, asphalt and concrete [62,63,64,65] Natural landscaping can also serve as a buffer against wind, rain and snow on colder or inclement days. Outdoor play spaces with significant natural features can provide the conditions that can allow children to play out longer and more often despite more increasing climate issues, and potentially serve as climate-resilient community hubs [63,66,67,68]. Outdoor playspaces which can be used extensively despite weather and climate issues may also be key environmental settings for reducing the spread of infectious disease, as was confirmed during the COVID-19 pandemic. 

### 1.8. Building the Evidence Base for Natural Playspace Design

Providing high-quality, diverse outdoor playspaces with abundant natural materials and opportunities for nature contact in community spaces such as parks and schools, may represent one of the best investments for supporting children’s healthy development and climate-resilience both in and beyond their early years. However, while a few studies to date suggest natural playspaces can change the character and improve the quality of outdoor play, little is still known about how the designed natural play environment affords specific and diverse forms of playful interactions. As well, very little research has worked to characterize outdoor play by more nuanced play types to help highlight the diversity of play supported (or not) by a given playspace. We need a larger and more robust evidence base which connects environmental features and conditions of these spaces to the promotion of positive play activities and outcomes for children.

One difficulty in building an evidence base which connects environmental features to high-quality play opportunities is the lack of a consistent approach or for assessing environmental support for diverse play. The recent development of a new typology of outdoor play types and associated observational tool [69,70] may provide the integrative framework by which we can both examine outdoor play behaviors and assess how and how well a playspace is supporting a full spectrum of play activity for a range of child users. Evaluating playspace environments through the lens of support provided for diverse types of outdoor play can help to identify key environmental supports, which in turn could inform advances in nature-rich playspace design. This study aims to significantly contribute to the development of a play type-focused evidence base by utilizing an intensive observational behavior mapping framework to capture and examine the environmental features and conditions of a large natural playscape which support diverse, developmentally supportive outdoor play activities among young children. First, we examine the characteristics, location, prevalence and inter-relatedness of outdoor play types afforded by the natural playscape; outdoor play type engagement will also be examined for differences by gender and age. Secondly, analyses aim to shore up the growing evidence that natural playspaces support both physically active and risky play behaviors, and connect these activities to specific play types as well as environmental features. Finally, we consider environmental engagement during play, specifically the potential scaffolding relationship between the presence of loose parts and varied topography and engagement in differing outdoor play types. 

## 2. Methods

### 2.1. Data Collection Methods

The study captured the outdoor play activities of children visiting *The Backyard* at the Santa Barbara Museum of Natural History (in Santa Barbara, CA, USA), a large naturalized playspace located adjacent to the museum surrounded by a canopy of native Coast Live Oak trees. *The Backyard* was designed to offer a play space where children can play freely in and with nature across a range of settings including a large creek, a mud kitchen and a build area (See Figure 1 for a labelled site plan and Figure 2, Figure 3, Figure 4, Figure 5, Figure 6 and Figure 7 for site photos).

Place-based behavior mapping protocols described in Cox, Loebach, and Little [69] were used in the collection of observed play events across *The Backyard* and included the recording of data such as outdoor play type, activity intensity, risk behaviors and environmental engagement. Data were collected by three observers over a seven-day period (5 weekdays, 2 weekend days) during a week in July 2019; at this time of year most schools in the region are on their summer holidays and *The Backyard* typically experiences peak use. The observers collected data during the hours that the playspace was open to visitors, from 10:00 am to 5:00 pm (less a one-hour break), resulting in a total of 126 observer hours. 

A basemap of *The Backyard* was created in Esri’s ArcMap by the authors and uploaded to an institutional ArcGIS Online Enterprise account. Behavioral and environmental data were input using ArcGIS Collector software on three Samsung Galaxy Tab S6 tablets. Data were collected off-line after basemaps were downloaded on to the tablets; recorded data were then uploaded to the same account at the end of each day. 

The site was divided into six discrete observation zones, and each observer spent 15 min in each zone, rotating through all six zones each round (One round of observation was completed every 90 min). Observers located themselves within each zone so that they could easily see and hear children as they played which enabled a more accurate and nuanced evaluation of the play episodes. Observers located themselves as unobtrusively as possible behind trees or amongst vegetation to avoid disrupting play or distracting children. 

Each zone was then scanned in a clockwise pattern. As the observer visually encountered a child as they were scanning the zone, they stopped to observe the individual for 15–20 s, and then recorded details of the observed behavior on the tablet. Each observation recorded the location of the individual geographically on the basemap, and then a pre-built form was used to collect multiple attributes related to the play episode, including age, gender and behavioral and environmental details. If the observer finished scanning the area before the 15-min period ended, the observer would scan again every five minutes, allowing for a maximum of three scans in each 15-min period. Attributes collected included demographic data, such as approximate age and gender; and numerous other attributes including physical activity level, risk behavior, environmental interactions, and an ‘open coded’ field where the observer wrote a short, but detailed, narrative of the observed play episode. (Attributes discussed in more detail below). 

Reliability across observers was reviewed at the beginning of each round by comparing data from observations on the same children. Discrepancies were discussed and noted to improve reliability for future observations.

### 2.2. Data Attributes Collected

The gender and relative age of each observed child was recorded, based on the observers’ best estimates. Gender was recorded as either male, female or unknown. Age group was categorized as 0–2 years (infant/toddler), 3 to 8 years (young children), 9 to 12 years (middle age children), and 13 to 17 years (adolescents). Only the data for children in the infant/toddler and young children categories (0–8 years) were included in this analysis. Due to difficulties estimating the age of child visitors, distinctions among young children were only drawn between those under and over 3 years to acknowledge the more limited physical and cognitive development of very young children compared to children beyond the toddler stage.

*Outdoor Play Type* was coded using the *Tool for Observing Play Outdoors* (TOPO), a typology and protocol which was specifically developed to better capture and understand children’s outdoor play behaviors [70]. Two outdoor play types (OPT) were recorded, whenever possible, for each observed play event. The TOPO includes nine primary categories of outdoor play: Physical play, Exploratory play, Imaginative play, Play with Rules, Bio play, Expressive play, Restorative play, Digital play, as well as Non-Play (See [69]). In its Expanded version, each primary OPT includes multiple sub-types of play for a total of 32 play type combinations. For this study, the Expanded version of the TOPO was used to categorize outdoor play behaviors. 

Physical activity was observed and coded using the Children’s Activity Rating Scale (CARS) which assesses energy expenditure for young children based on observed behaviors [71]. The scale includes five categories from 1 (stationary—no movement) up to 5 (translocation—fast, very fast/strenuous movement). 

*Risk-related play behaviors* were assessed via observation using the categories developed by Little [32]. Little’s risk behavior typology includes nine categories: risk avoidance, exploratory risk appraisal, very low/no risk, low risk (positive), low risk (negative), moderate risk (positive), moderate risk (negative), high risk (positive), and High risk (negative). Positive risk includes activities where the child is unlikely to be injured and negative risk taking includes inappropriate or unreasonably harmful behaviors [32]. Little’s risk typology is relational, that is, similar activities may be rated with different risk levels depending on the attitudes and abilities of the observed child. 

*Environmental interaction* data included observations about the involvement of fixed or loose parts as part of the play event. Up to three environmental interactions could be coded for each observation, and were based on whether the item was fixed (immovable) or loose (movable) and whether or not the item was in its natural form (e.g., pinecone, stick, water, mud) or manufactured (e.g., spoon, wooden boat, bowl). 

All areas within the site were categorized by one of four topographies: little to no slope, moderate slope, steep slope and uneven ground (See Figure 12 for topography designations).

All protocols for observing and documenting the play activities of visiting children were approved by the Institutional Review Boards of both the University of Kentucky and Cornell University, as well as the administration of the Santa Barbara Museum of Natural History.

### 2.3. Data Analysis Methods

Once data were downloaded from ArcGIS Online account, they were put into ArcGIS Pro v2 (ESRI). Data were sorted and selected in various ways and displayed using this software. Basemaps were evaluated along with site photography and observer’s experience to determine slope (categories: low, moderate, steep, and uneven terrain). All point and heat maps were created in ArcGIS Pro v2.

Observational data were also exported from ArcGIS into both Microsoft Excel (for Microsoft 360) and Stata v17 for further quantitative analysis. Stata was used to produce two-way tables comparing all categorical variables; relationships between categorical variables were calculated using Fisher’s Exact Test and further interpreted using Cramer’s V from Pearson chi-squared analyses.

## 3. Results

Over the course of seven observation days in *The Backyard*, a total of 693 play events were captured for children 8 years old and younger. Male children comprised close to 60% of the observed players, while females represented just over 40% (See Table 1). The vast majority were children between 3 and 8 years old (87.6%) but 12.4% of observed play events involved children 2 years or younger. See Figure 8 for Backyard map of all observed play events.

### 3.1. Overview of Backyard Play Behaviors

Physical Play and Exploratory Play were each involved in more than 60% of observed play events [recall that the TOPO protocol allows each play event to be categorized by up to two outdoor play types] (See Table 1). Almost 10% of play events involved an Imaginary Play component, and about 7% included Play with Rules. Bio Play, Expressive Play and Restorative Play were each identified as part of approximately 5% of play events. Digital Play was extremely low, with only one observed event. Non-Play activities such as nutrition breaks, transitioning, and self-care, were the third highest proportion of observed activities, involved in almost 18% of observations.

*Play Activity Intensity:* Observations of activity intensity using the CARS revealed that the playspace was supporting a wide range of intensities from stationary to quick movements. The majority of the play activities fell into the middle range in terms of intensity; each stationary with limb movements, slow and moderate levels represented a quarter to a third of all play activity (See Table 1; Figure 9). Stationary activities comprised just over 10%, while the most intensive level, quick movements only represented about 5% of all observed play. However, when condensed to stationary, slow and moderate-quick levels, each represented about one-third of all play activities.

*Risk-Related Behaviors:* Using all of Little’s risk categories, observations indicated that close to half of play activities involve no to very low levels of risk. However, more than half of the play events (52.8%) exhibited characteristics of positive behaviors, particularly low positive risk (34.5%) and moderate positive risk (17.6%); less than 1% were categorized as involving high positive risk (See Table 1; Figure 10). In contrast, very few play events involved any level of negative risk; of 693 play events, only 5 (less than 1%) had elements of negative risk.

***Environmental Engagement:*** Examining the relative engagement of natural and manufactured elements in play activities we find that 83.6% involved natural elements, either fixed or loose parts, while just over 70% involved manufactured elements (See Table 1).

*Role of Loose Parts:* Observations revealed that 72% of all play activity involved loose parts found within the playspace; with the exception of Expressive and Non-play activities, more than 50% of activities for each play type involved loose parts (See Table 1; Figure 11). Observed play often involved multiple loose parts, both natural (such as mulch, leaves, water and acorns) and manufactured (toy boats, bowls, spoons, binoculars). Children used natural loose parts in 63% of play events, and manufactured loose parts in 45% of play activities.

*Topography:* The entire Backyard space was categorized by four topography categories (See Figure 12). Half (50%) of all observed play took place in settings with little to no slope; 43.4% of play activity took place in areas with uneven surfaces, such as the boulder-lined and -filled creek (See Table 1; Figure 13). Play on moderately or steeply sloped areas comprised just over 6% of all activity.

### 3.2. Overview of Outdoor Play Types and Associated Conditions

All of the nine primary TOPO outdoor play types were observed in *The Backyard*, but some much more prevalently than others. Results associated with each play type are outlined below.

***Physical Play***: More than 60% of all play events involved a Physical play component (See Table 2). The majority of Physical play activities (60.2%) also involved an Exploratory element; almost a quarter of Physical play (20.3%) involved more than one Physical play subtype, and 7.5% was paired with Play with Rules. Gross motor play comprised the largest proportion of Physical play, observed in over 70% of Physical activities, and 44% of all observed play events, and largely centered on children climbing up and down the large boulders in and along the upper and lower creek area, lifting bamboo poles in the build area to construct teepees and forts, and running down ramps or slopes (See Figure 14). Gross motor play was often paired with vestibular components (20.3% of all Physical play, and involved in 13% of all observed play), where children might be both climbing boulders and logs, then balancing their way across them. Fine Motor play, such as manipulating boats to sail or race down the creek, or stirring bowls of mulch in the mud kitchen, was evident in almost 30% of all Physical play and 17% of all observed play; often paired with Exploratory or Imaginative play, Fine Motor play was observed most often in the creek and mud kitchen where there were significant loose parts (See Figure 14).

***Exploratory Play****:* Activities with an Exploratory component were the second most observed outdoor play type; in addition to being predominately paired with Physical play (62.4%), it was also often observed in combination with Imaginative (14.1%) and Bio play (8.0%). Exploratory-Active play was the most common subtype observed, involved in 66% of all Exploratory activities and 40% of all play events, and was heavily centered on experimenting with boats in the water (including ways to manipulate the boat or the creek environment to change its trajectory down the creek), and mud kitchen activities such as filling or stirring bowls full of mud, water, mulch and other natural loose parts (See Figure 15). Exploratory-Sensory play (in 26% of Exploratory activities, 16% of all play events) was also prominent and focused on exploring the properties of water or mud, or looking for or examining the sensory features of loose parts such as water, leaves, mulch and flowers. Sensory play was observed in diverse locations across the site, particularly where plants or other natural matter was present. Constructive play, observed in 10% of all Exploratory activities, was largely observed in the build area where children would build structures using the bamboo poles present, and in the creek where they built dams in various parts of the creek (though this was not allowed due to the sensitivity of the creek water system and was discouraged or dismantled fairly quickly by staff).

***Imaginative Play****:* Almost 10% of all play observed clearly involved Imaginative play elements (See Table 2), and of these, 82.8% were combined with Exploratory play behaviors such as manipulating the environment to simulate cooking or playing ‘house’ in the mud kitchen, or pretending to be, for example, a boat captain while maneuvering a toy boat down the creek. Just over 15% were paired with Physical play, primarily Fine Motor movements. Many play activities were characterized as Imaginative-Fantasy (29.9%), where children were heard pretending to be superheroes, wizards, monsters or animals; these Fantasy play activities were observed in diverse locations across the site (See Figure 16). Symbolic play comprised 13.4% of all Imaginative play, and 1.3% of all play events; often coded Symbolic when there was not sufficient evidence of other Imaginative sub-types, this form of play was observed in diverse locations across the site.

***Play with Rules:*** A small portion (6.8%) of observed play was in part characterized as Play with Rules, where children agree to play with a given or negotiated set of rules (See Table 2). Play with Rules was also heavily associated with Physical play (69%), but also more than a quarter involved Exploratory components. Nearly all Play with Rules (91.5%) was of the Organic subtype, where children have developed their own unique rules to govern a cooperative play activity. While made up games or obstacle course runs were observed, the majority of this subtype involved children developing rules for racing boats down the creek. About 9% of rule-based play was Conventional, reflecting commonly known games such as *Hide N Seek* or *Duck*, *Duck*, *Goose*; many of these games were observed in or around the stump circle, but they also appeared in diverse locations across the site (See Figure 17).

***Bio Play:*** A newly introduced outdoor play type, Bio play, comprised 5.6% of all observed play events, and the vast majority (85.7%) of these activities were observed in conjunction with Exploratory play behaviors, particularly Exploratory-Sensory (See Table 2; Appendix A). The largest proportion of Bio play involved Wildlife interactions (89.7%) and centered primarily around the large birds and reptiles which were present in exhibit buildings on the perimeter of the playspace but also regularly brought out into the center open space for exhibition by staff naturalists and Audubon Society volunteers. However, children were also recorded observing or following butterflies, wild birds, and insects in diverse locations across *The Backyard*. About 10% of Bio play was focused on plant material such as feeling soft plant leaves or smelling flowers and was observed in various locations. No play was observed which involved Bio-Care activities.

***Expressive Play:*** Also associated with about 5% of play events were Expressive play elements (See Table 2; Appendix A). The largest portion of these playful interactions involved social conversation (77.1%) with other children or else adults such as parents or staff; as much of the Conversation activities took place while being seated with others or eating lunch together, Expressive play was often found alongside a Non-play element (56.7%). Some Expressive play took the form of Performance (8.6%), such as singing or dancing for others, which was reflected in the common pairing of Expressive play with Physical play activities (23.3%). Expressive-Language play was also occasionally observed (17.1% of Expressive play; 1% of all play) and manifested as children singing or talking to themselves, or telling stories and jokes to one another. The high proportion of Expressive-Conversation activities resulted in some activity density around benches in the paleo area and other locations, but was otherwise well distributed across the site (See Appendix A).

***Restorative Play:*** Another new outdoor play type, Restorative play was associated with about 5% of play events observed (See Table 2). The majority was categorized as Onlooking behaviors (58.3% of all Restorative, 3% of all play), where children would sit away from others, often on the edges of play settings, and quietly observing other children or patrons in the space (See Appendix A). Much of this time was also spent observing the landscape around them or idly playing with environmental elements such as leaves or water, and so was commonly paired with Exploratory play (43.3%). Resting was another prominent subtype (47.2% of all Restorative, 2.5% of all play), and typically involved children sitting on boulders, stumps or benches, or else lying down in shady spots under trees or bamboo structures. Restorative play activities were widely distributed across *The Backyard*.

***Digital Play:*** Only one observed play event involved digital play, where a child was sitting on a bench playing on a cell phone (See Appendix A). The lack of observations prevent the ability to make associations with other play types or prevalent environmental locations or features.

***Non-play:*** Almost 18% of observed activities involved a Non-play component (See Table 2). The largest proportion of Non-play was characterized as Transition behaviors (45.9% of all Non-play, 8.1% of all play), where a child is moving between play settings or else entering or exiting the playspace. These transitional behaviors are also commonly observed in combination with a child also visually engaging with the landscape around them, and so are often paired with Physical-Gross Motor (31.5%) or Exploratory-Sensory (30.1%) play behaviors. Transitions between or in/out of play settings where a child is accompanied by other children or adults often includes Conversation, and so is frequently seen alongside Expressive (23.3%) behaviors. As part of an overall play cycle, children often took time out for Nutrition breaks (34.4% of Non-play) or Self-care activities (15.6% of Non-play) such as tying a shoelace or changing out of wet clothes. A small number of Non-play behaviors included a child observed in distress (1.6%), crying or arguing with a peer or parent, but no examples of Aggression behaviors were observed. Non-play activities were not concentrated in any particular area but rather were observed across the playspace.

### 3.3. Demographic and Behavioral Associations with Outdoor Play Activities

Note that results related to Digital play will not be reported in this section due to the very low level of observations.

***Gender*:** Among male children, Physical play was the most observed OPT (39.9%) followed by Exploratory (35.5%), whereas Female children were observed most in Exploratory play (33.7%) followed by Physical play (31.0.5%) (See Appendix A). A Fisher’s exact test indicated a significant association between males and Physical play (*p* = 0.000), but only indicates a fairly low level of association (Cramer’s V = 0.16) (See Appendix A). Play with Rules also showed a significant but low association with males (*p* = 0.001; Cramer’s V = 0.12). Significant but low associations were found between females and each Expressive, Restorative and Non-play types. However, overall there were no strong associations for any OPT with a particular gender. 

***Age Group:*** While the majority of child players were in the 3 to 8 year old range, there were still differences in levels of engagement in various OPT between the two age groups. Children 3 to 8 years old demonstrated equal levels of engagement in Physical and Exploratory play, each involved in about 35% of activities (See Appendix A). Children 2 years and under engaged in a higher proportion of Physical play (45.8%) than older children but still exhibited a fairly high proportion of Exploratory activity (36.1%). Neither Imaginative play or Play with Rules was observed at all among the younger cohort, whom had slightly higher levels of Non-play. Older children exhibited slightly higher engagement in Bio, Expressive and Restorative play. There were significant but ultimately low associations between the older age group and both Imaginative play (*p* = 0.001; Cramer’s V = 0.12) and Play with Rules (*p* = 0.008; Cramer’s V = 0.10) (See Appendix A). There was also a significant but low association between the younger cohort and Physical play (p=0.004; Cramer’s V = 0.11). There were no highly significant differences in OPT engagement across the two age groups. 

***Activity Intensity:*** Using the three condensed CARS categories, the proportion comprising each OPT was calculated (See Appendix A; Figure 2). Play with Rules (48.9%) and Physical play (42.2%) exhibited the two highest proportions of moderate-vigorous activity. Exploratory, Imaginative, Expressive and Non-play each involved 20% to 30% of this higher intensity activity. Restorative (83.3%), Bio (79.5%) and Expressive (68.6%) play exhibited the highest proportions of stationary activity. Exploratory, Imaginative, and Non-play each involved 40% to 50% stationary activity. Physical play was significantly (*p* = 0.000) and moderately associated (Cramer’s V = 0.43) with moderate-vigorous activity while Exploratory play was significantly (*p* = 0.000) and moderately associated (Cramer’s V = 0.30) with stationary and slow play activities (See Appendix A). There were significant but ultimately low associations between moderate-vigorous activity and Play with Rules, and between stationary activity and each Bio, Expressive, Restorative and Non-play.

***Risk Behaviors*:** Using the three condensed Risk Behavior categories, the proportion comprising each OPT was calculated (See Appendix A; Figure 3). Bio play (89.7%) and Non-play (84.4%) exhibited the highest proportion of no to low risk behaviors; Imaginative, Expressive and Restorative each involved 70% to 75% no/low risk behaviors. Play with Rules (83.3%) and Physical play (71.9%) each demonstrated high proportions of positive risk activities. Several OPT including Exploratory, Imaginative, Expressive and Restorative also involved fairly high proportion of positive risk, from 25% to 50%. Negative risk behaviors were only observed in association with Physical and Exploratory play, but comprised approximately 1% of less of all activities. Imaginative, Bio, Expressive, and Restorative play all had significant but low associations with no to low risk behaviors, and Play with Rules had a significant but low association with positive risk. However, Physical play was significantly (*p* = 0.000) and strongly associated (Cramer’s V = 0.51) with positive risk behaviors.

### 3.4. Environmental Associations with Outdoor Play Activities

***Loose Parts:*** All OPT with the exception of Expressive and Non-play showed loose parts engagement in at least half of the observed play activities (See Appendix A; Figure 4). Exploratory (89.9%), Imaginative (88.1%) and Play with Rules (83.0%) each exhibited extremely high levels of loose parts involvement. Exploratory play was shown to be significantly (*p* = 0.000) and strongly associated (Cramer’s V = 0.49) with loose parts, and significant but low associations were also found with Imaginative play activities (See Table 3). Expressive, Restorative and Non-play had significant but low associations with no use of loose parts.

***Natural Loose Parts****:* Examining use of only naturally found loose parts (NLP), Exploratory (80.5%) and Imaginative (83.6%) play activities exhibited the highest proportions; Physical, Bio and Play with Rules each had 60% to 75% natural loose parts involvement (See Appendix A). Exploratory play was significantly (*p* = 0.000) and moderately strongly associated (Cramer’s V = 0.46) with NLP (See Table 3); Imaginative play had significant but low associations with natural loose parts. Expressive and Restorative play each had significant but low associations with no use of NLP, while Non-Play was moderately associated with no use of natural loose parts.

***Manufactured Loose Parts****:* Manufactured loose parts (MLP) were less involved in observed play than natural loose parts; Play with Rules (78.7%) and Imaginative (73.1%) play exhibited the highest proportions, while MLP were observed in about half of Physical and Exploratory play activities (See Appendix A). The remainder of the OPTs showed less than 20% involved of manufactured loose parts. Use of MLP was significantly but only lowly associated with Physical, Exploratory, Imaginative and Play with Rules (See Table 3). Bio, Expressive, Restorative and Non-play each showed significant but low associations with use of no MLP as part of play activities. 

***Topography****:* For most OPTs, more than 50% of play activities were observed in areas with little to no slope; more than 60% of Imaginative, Bio, Restorative and Non-play activities took place in fairly flat areas (See Appendix A; Figure 5). Play with Rules (80.9%) and Physical (53.5%) play saw large proportions of activity taking place on uneven surfaces. Physical play was responsible for the highest proportion of all play activities on moderate (35.7%) and steep (34.8%) slopes, as well as uneven (45.5%) surfaces. Physical play and Play with Rules showed significant but low associations with uneven surfaces (See Table 3). Exploratory, Imaginative, Bio and Restorative play each had strong but low associations with areas with no/low and moderate slopes. 

## 4. Discussion

Analysis of play events in *The Backyard* revealed key details regarding the diversity and prevalence of outdoor play activities observed, as well as the environmental settings and features which afforded various outdoor play types.

*The Backyard* provided opportunities not only for all types of outdoor play, but for diverse sub-types as well, reinforcing claims by other studies that natural playspaces tend to increase the diversity of play behaviors observed [21,24,72]. 

As with a few other studies of natural playgrounds [4,46,73,74]. Physical play was the most prominent outdoor play type observed, involved in more than 60% of all observed play. Physical-gross motor was in turn the most prevalent subtype and manifested largely through climbing up and down the large boulders and stumps located throughout the site, building structures with large loose parts, as well as running down or up the moderate to steep slopes crisscrossing the site. Many of these activities required the need to balance successfully and so gross motor activities were often observed together with Physical-vestibular play. Play involving fine motor skills also made up a large component of many activities, particularly as children moved and manipulated small loose parts such as toy boats in the creek and spoons and bowls in the mud kitchen.

Similar to previous studies, Physical play was supported in numerous ways through both the fixed and loose environmental features on the site [25,26,40,75]. The presence of large, fixed features such as boulders, stumps and logs of varying sizes and heights set in close proximity to one another supported both climbing up and jumping across for all ages. Areas with uneven surface topography, such as the boulder-filled creek, were also significantly associated with Physical play; combined with moderate and steep slopes and ramps, *The Backyard* provided ample and challenging opportunities to run up and down parts of the site at significant speed, while requiring some careful negotiation of changing features. The strongly significant association between Physical play and both higher intensity activity and positive risk also supports the contention that environmental provision for Physical play can support healthy physical activity among children, as well as the risky play which can scaffold the advancement of skills and greater environmental competence [35,42,52]. These strong associations also demonstrate that *The Backyard* features provided diverse opportunities for Physical play appropriate to young children of all ages and skills levels.

All subtypes of Physical play were also substantially supported by the presence of numerous loose parts such as boats, spoons and bowls along with manipulable, fluid materials such as water, dirt and mud. While not always intuitively considered as scaffolds for Physical play, the close and plentiful presence of loose parts in *The Backyard* were key supports for Physical play activities, echoing findings from a 2012 review which found that the presence of portable play equipment was positively correlated with more physically active play [76]. The prevalence of Physical play and its strong association with more active behaviors reinforces other literature which ties outdoor play to higher levels of physical activity and physical literacy skills among young children [4,77,78]. We must consider though that this strong finding across the literature may in part be explained by the fact that we predominately design outdoor spaces for Physical play, with little focus paid to supporting diverse forms of play. While these findings provide some key insights about which environmental features and conditions we can integrate into outdoor playspace design to continue to encourage Physical play, we also advocate for intentionally designing to support other outdoor play types as well.

A few previous studies have distinguished between different types of outdoor play observed among young children in a playspace [24,25,74], but none to date have examined the inter-relatedness of these play types. *Backyard* observations revealed that Physical play was strongly paired with Exploratory play activities; these outdoor play types were co-present nearly two-thirds of the time. In particular, Physical-gross motor was often observed with either Exploratory-active or -constructive play, as children manipulated, experimented with or built with features in the playspace environment. Physical-fine motor was often observed together with Exploratory-sensory or -active play, when these more nuanced movements were used to explore the sensory features of the environment such as picking up a rock or acorn to examine or scooping up water or mulch into a cup and pouring it out elsewhere. Not surprisingly, Physical Play was also often observed alongside Play with Rules as these activities often involved physically active chasing or racing games. While other research has tied open, flat expanses in play spaces to prompting physically active games [22,24,25], this study clearly demonstrates that these conditions are not always necessary; the craggy, boulder-strewn creek also prompted physically active games. The strong inter-relatedness between outdoor play types across this study also suggests that focusing on environmental features which support a given outdoor play type will likely have positive spillover effects for other play types. For example, aiming to provide environments which support the more active end of Physical play activities which are often combined with Play with Rules or Exploratory-active, as well as the quieter or less intensive physical play associated with Imaginative and Exploratory-sensory play, will provide a greater and more appealing range of Physical play activities which not only support children’s full physical development but their cognitive and creative development as well. 

Exploratory play was also highly prevalent in *The Backyard*. Exploratory-active made up the majority of these activities and nearly always involved the use and manipulation of loose parts in the environment, particularly the maneuvering of boats down the challenges of the boulder-strewn creek or the handling of tools and natural materials such as mulch, leaves, dirt and water to make ‘soup’ or ‘potions’. The prominent manipulation of these environmental materials for pretend scenarios meant that Exploratory-active and -constructive activities were often observed in concert with diverse Imaginative play activities. Exploratory-sensory activities were also sometimes paired with Bio play, as children actively used their senses to look for and examine living elements in the playspace, with a particular focus on wildlife—either the caged birds of prey prominently displayed or observing birds, butterflies and insects that lived in and around the playspace. The presence of natural vegetation throughout the site provided the habitat for such wildlife which was key to the provision of these exploratory activities, along with promoting myriad opportunities for direct contact with nature. 

While earlier research has focused primarily on associations between Physical play (alternatively categorized as Functional play) and activity intensity, our findings also demonstrated that Exploratory play activities were moderately associated with the lower intensity physical activities that can advance fine motor skills. While some Exploratory play was more physically active, much of the activity observed involved intense but slower-paced engagement with or playful examination of features in the environment, and tended to be observed in discrete, quieter settings with little to no slope set further away from the action, such as the mud kitchen, and the paleo and build areas. 

Similar to other studies [47,73,74], Exploratory play was also strongly associated with loose parts in general, and natural loose parts in particular. Manufactured loose parts still featured prominently as key supports for Exploratory play but were not as strongly tied. The availability of suitable and fairly plentiful loose parts which were light enough to manipulate, either intentionally provided or naturally present in the space, and located in close proximity to key environmental settings such as the creek, mud kitchen and build area, substantially scaffolded children’s engagement in Exploratory play in *The Backyard*. The presence of water was also key to a large proportion of Exploratory activities. The availability of moving water in the creek, combined with toy boats and smaller loose rocks which could be moved to create small dams or rapids, served to be one of the most attractive affordances provided by the site. In the mud kitchen, where children could access water at will through a child-scaled faucet, its presence significantly expanded the playful opportunities available to children, including its combination with dirt to create mud. 

In addition to confirming the symbiotic relationship between loose parts provision within or near to fixed play infrastructure, these observations again reinforce the value of an OPT lens for assessing environmental provision not only for more active exploration, but for the more passive, contemplative forms of Exploratory, Bio and Restorative play which require settings which are somewhat removed from higher energy play and which supply plentiful manipulable parts and materials. 

While not as commonly observed, Imaginative play still comprised about 10% of all observed play, and was predominately co-present with Exploratory play activities. Not surprisingly, Imaginative play was only observed among players in the 3 to 8 year cohort; this is not to say that younger children were not involved in some form of Imaginative activities, but observing this form of play can be difficult when children do not vocalize their pretend play behaviors or clearly show actions of Symbolic play such as holding a short stick up to their ear and pretending to talk on a ‘phone’. However, during child development imaginative activities only begin to manifest between one and three years of age [79,80], which is reflected in these findings. 

Imaginative activities largely took two forms: semi-stationary Sociodramatic play that stayed largely within a specific setting such as the mud kitchen, and the more active Fantasy play and games, such as playing superheroes chasing villains or piloting a runaway ship down the ‘rapids’ in the creek, which tended to move more broadly within and across different settings. As such, Imaginative play behaviors reflected all levels of activity intensity. The more active Imaginative play activities along the creek or crisscrossing the whole site, often exhibited risky elements, which is evidenced by the significant though somewhat low association with positive risk. Sociodramatic play, the most prevalent Imaginative subtype, was particularly supported by the presence of play prompts such as spoons and bowls, toy boats and binoculars, in conjunction with fluid materials such as water and mud. As found in other studies, the availability of building materials or else fort-like structures previously built by other children, also served in some cases as a play base or prompt [47,49]. Imaginative play which could involve significant movement across the site, such as running away from a monster, were aided and enhanced by clear pathways and undulating topography. While previous studies have linked risky play or active play with environmental features such as uneven surfaces and hard surfaced pathways [25,74], none are known to have associated these with Imaginative play activities. *The Backyard*, being a large space, provided not only enough room for diverse forms of Imaginative play, but distinct settings and pathways which allowed for both more active and more passive activities to take place without conflict. In addition to making this environment-behavior connection, these findings again highlight the value of assessing a play environment in terms of the play types it supports. 

Collectively Bio, Restorative and Expressive activities made up over 15% of observed play and while not as prominent as other play types are a strong reminder of the appeal of these quieter, restful and more creative activities for children. The largest environmental support for Bio play was the presence of larger birds and reptiles kept on site, which were frequently brought out for exhibition and examination by informed naturalists and handlers. However, the nature-rich character of the playspace along with wildlife structures such as bird and butterfly houses, also attracted or provided habitat for wild birds, butterflies and insects, which in turn increased the opportunity for direct engagement with living things in the space.

Bio, Restorative and Expressive play were also associated with lower intensity activity levels, and each was also associated with play involving lower risk activities. Spaces which support these quieter, lower energy and low risk activities can provide a welcome change or break from risky, active play opportunities undertaken in other areas as part of an overall play cycle. Both Restorative and Expressive play were served by some similar environmental supports; the majority of these activities took place in areas with little to no slope, particularly where there were opportunities to sit—either on one’s own or with others—separated somewhat from the action. Plentiful seating opportunities were provided by benches, tree stumps as well as the boulders strewn throughout the space. Some observed Restorative activities took place from seating perches that offered prospect, that is, the chance to look out from high points or from the ‘edges’ over the play space and watch the activity happening down below. Small structures and natural nooks provided quiet, shady ‘away’ places for one or a just a few children to sit, chat, and/or retreat. Bio and Restorative play were also aided by the substantial presence of mature trees; in addition to serving as sources of leaves and habitat to attract wildlife, the trees provided significant shade in the area which made it easier to linger and play for longer periods, despite the summer heat.

The moderate connection between both Expressive and Restorative play activity and the lack of use of loose parts is not surprising as these forms of behaviors tend to have a high social component and do not necessarily rely on play prompts available within the play space. However, more diverse loose parts might have better served more varied forms of Expressive play. While we might have expected Bio play to be associated with loose parts, the lack of relationship is likely due to the prevalence of more structured but passive interaction with birds and reptiles present on site rather than more organic engagement with naturally occurring plants and wildlife. Without the wildlife program we might expect to see a stronger relationship to loose parts involvement. 

One objective of this study was to consider whether there were significant differences in play engagement by gender or age cohort. Several of the play types showed that engagement was slightly higher among female than male children, specifically Expressive, Restorative and Non-play activities. More of the boys observed were engaging in more physically active play, while girls were tending to choose quieter, less intensive activities which also included significant amounts of social conversation as well as self-care activities such as resting and taking nutrition breaks. It may not be that females were less interested in Physical play but rather were involved in a greater diversity of play types and so more physical activities featured less prominently. However, similar proportions of male and female children engaged in Bio play, particularly interacting with wildlife, suggesting that interaction with the natural world is appealing to children of any gender. Despite some small differences in play type preferences, these differences were not strongly significant, reflecting little overall gendered difference in play type engagement. 

Considering differences by age we found that while children under 2 years engaged in a higher proportion of Physical play than the older cohort, this difference was not strongly significant; Physical play comprised a substantial part of the play activities of all children 8 years and under. Children under 2 years were not observed to engage in any Imaginative play or Play with Rules, but these findings are in keeping with the developmental abilities of very young children. No other strongly significant differences between the two age cohorts were found in types of play activities observed, suggesting not only that all age groups are interested in diverse forms of outdoor play but that *The Backyard* provided features supportive of diverse play appropriate to all children under 8 years old. However, the grouping of all children between 3 and 8 years may be hiding some potential activity differences within this sub cohort. 

This investigation also sought to examine the role of environmental features such as loose parts and varying topography in supporting or prompting different types of outdoor play. The very strong relationship between loose parts, especially natural loose parts, and numerous outdoor play types shores up the growing body of evidence which highlight loose parts as key contributors to enriched play, and the value of incorporating loose parts in outdoor play spaces wherever possible. 

The role of varied topographies has largely gone unexamined to date in the literature, but these findings, along with a few other studies, suggest that this environmental feature could be a significant prompt and scaffold for outdoor play. Additional work to confirm this relationship would be a significant contribution to the scholarship and provide a clear rationale for leveraging this feature of outdoor playspaces to enrich their play affordances. 

Overall, the analysis of observed outdoor play activities in the nature playscape of *The Backyard* yielded a number of key insights that can help to inform the design or improvement of outdoor playspaces. First, findings illustrate that plentiful and diverse play settings on the site, coupled with changing topography, engaging loose parts, and separation between higher and lower intensity activities, are capable of supporting diverse outdoor play types for young children under 8 years but also provided multiple locations supportive of each play type. The environmental features of the site, including diverse topography and varying sized boulders and stumps afforded opportunities for all levels of activity intensity as well as graduated challenges and positive risk opportunities regardless of age. The extremely high involvement of both natural and manufactured loose parts in most play types emphasizes the key role that these features perform in scaffolding and enriching diverse forms of play, echoing findings from earlier studies. The agile and non-prescriptive character of natural loose parts in particular appears to encourage children to curate their own play experiences while allowing for play behaviors to easily evolve or shift [39,46,73]. The role of water in *The Backyard* play should also be highlighted. The active movement of water down the length of the creek substantially increased the play opportunity and sensory experiences provided by this setting; this living, changing system was responsible not only for the most play activity but for sustained play engagement. The faucet in the mud kitchen which allowed children to access water at will, and utilize it as its own play element and add it to dirt to form the incredibly fun medium of mud, was a substantial scaffold to more imaginative and sustained play among a very diverse cohort of children. While water systems can be complex and require more supervision and maintenance, the significant increase in play value that they provide should make them worthy of consideration whenever possible.

A final note too about risky play, which tends to receive less attention in the design of play spaces than provision for physically active play despite that fact that both have been associated with developmental benefits [33,35]. While always attempting to eliminate any hazards which may present, playspaces should aim to provide opportunities for children to safely undertake and conquer positive challenges during their play activities to promote skill development and environmental competence. However, these positive risks are relational; that is, what presents a challenge for a young child still developing physical skills, such as climbing a small boulder, may no longer represent a challenge for an older, bigger child. The risky play opportunity for this child may manifest rather in a series of small boulders in close proximity which allow them to enhance their skills by jumping across and balancing from one boulder to the next. Playspaces which can provide such diverse and ‘graduated’ challenges can provide risky play opportunities for a broader range of child players. The environment of *The Backyard* provided significant positive risk opportunities for children of all ages and abilities in this cohort. Observations also revealed very low occurrences of negative risk behaviors, approximately 1% of all observed play; this finding demonstrates not only that children are drawn to risky play but that despite opportunities provided by the environment supporting higher levels of risk for those with the appropriate skills, most children will largely police their own behaviors, taking on only those low or moderate risks which will advance their development without risking serious harm or injury. Findings from this study support the theory that playspaces which provide ample prospects for positive risk for a range of skill levels can support more appealing play opportunities which can help to advance child development.

While the site substantially supported high-quality outdoor play, there were some features and conditions which might have also served as barriers to or limiters of play. The density of activity in the creek and mud kitchen areas reveal that the majority of appealing play affordances (such as the bulk of the loose parts) on site might have been clustered in these areas, suggesting that the design of less popular settings could be enriched to disperse play opportunities more broadly across the site. The substantial separation of the build area from the rest of the play space may have also limited opportunities for play to flow more naturally between this and other play settings; the separation also prevented the loose parts available in the build area, such as pine cones and tree cookies, from being utilized in other spaces, which would have further increased the richness of available loose parts. Some ‘environmental science tools’ such as binoculars and paleo exploration kits were available to kids to use within the space, but to control their use they were housed in or near the clubhouse where they could be more easily distributed by staff; the consequence was that these instruments were not necessarily near to the areas where the children might use them, and seemed to substantially limit their use. Therefore, proximity of loose parts to play settings appears to be in part a predictor of use. Finally, while the creek water system was one of the largest draws for children, the particular system used could not tolerate a lot of natural material added to it; children were therefore restricted from using sticks and other materials to create more substantial dams or ‘rapids’ for their boats to navigate (a staff member was usually nearby to curb such activities). They also could not test how well different natural materials floated or made their way down the creek. The limitations of the water system limited even more diverse interactions and learnings from this play setting. 

## 5. Strengths and Limitations

A key strength of the study was the rich and detailed field data provided by the observation behavior mapping which allowed for specific play activities and characteristics to be directly tied to environmental features and conditions. This connection is vital to understanding how play environments support (or not) diverse forms of play, which is necessary to advancing the field of playspace design. Framing the assessment through the new TOPO outdoor play typology also strengthened the consistency and transferability of the findings, and demonstrated its value in serving as a common research frame for outdoor playspace evaluations moving forward. Many outdoor play studies to date have only considered a single contributor or outcome, such as loose parts provision or level of physical activity, in relation to a play environment. The investigation of relationships with multiple behavioral and environmental variables in this study provides a more holistic assessment of a play environment, and sets up the ability to consider more complex interactions among study variables. Overall the study provides increased evidence of supportive environment features which can inform the development of new play environments or the assessment and improvement of existing playspaces.

A few methodological and resource issues led to some limitations for the study and the applicability of its findings. While in part necessary to increase the reliability of field observations, the lack of further distinction by age among the children aged 3 to 8 years prevented a more nuanced understanding of similarities and differences within this sub cohort. Future work will aim to further divide the age boundaries to assist with more refined evaluations. Another limitation was the lack of a site survey or detailed vegetation plan which would have allowed us to examine the impact of topography and vegetation in a more detailed way. As well, while the study considered numerous behavioral and environmental variables in relation to outdoor play activities, interaction effects between these variables have not yet been investigated. Future studies will work to investigate potential moderating effects. Finally, while a significant number of observation hours were logged by behavior mapping research standards, resource limitations meant observations were limited to a single week of the year; activity patterns and relationships which emerged could be strengthened by additional weeks of observation, including observations at difference times of year.

## 6. Conclusions

This study set out to examine the types and characteristics of outdoor play behaviors supported by the environmental features of a large natural playspace, *The Backyard* at the *Santa Barbara Museum of Natural History* in California. Analyses confirmed that the diverse, highly naturalized setting supported almost all outdoor play types for both age groups and genders, and offered multiple opportunities across the site for engagement in each form of play. *The Backyard* assessment confirmed that such natural playspaces can equally afford a full range of activity intensities as well as substantial opportunities for positive risk-taking behaviors. The highly influential role of environmental features such as loose parts and varied topography demonstrated that play novelty and value can be significantly increased through fairly simple and economical environmental alterations. Deliberately designing natural playspaces to support all outdoor play types can also help a space to support a greater spectrum of young children’s play and interaction preferences, allowing children to choose to engage in the type of play and degree of interaction which suits them best at a given time and stage. 

This study also confirmed the efficacy of an observational behavior mapping framework, particularly when combined with the *Tool for Observing Play Outdoors*, for capturing key characteristics and nuances of children’s outdoor play behaviors and for examining the role of the play environment. While this study significantly contributes to the critical evidence base need to design more supportive play environments, additional robust research to further shore up the connections between playspace design and positive play behaviors will be highly valuable and will help to inform clearer guidelines for the development of high-quality outdoor playspaces for children of all ages. Increasing the quality of outdoor playspaces for young children, particularly those that provide positive exposure and interaction with nature, may also serve as a mechanism for prompting pro-environmental attitudes and behaviors across the life course, and provide public green spaces which can serve as resilient community resources as we face a changing climate.

## Figures and Tables

**Figure 1 ijerph-19-12661-f001:**
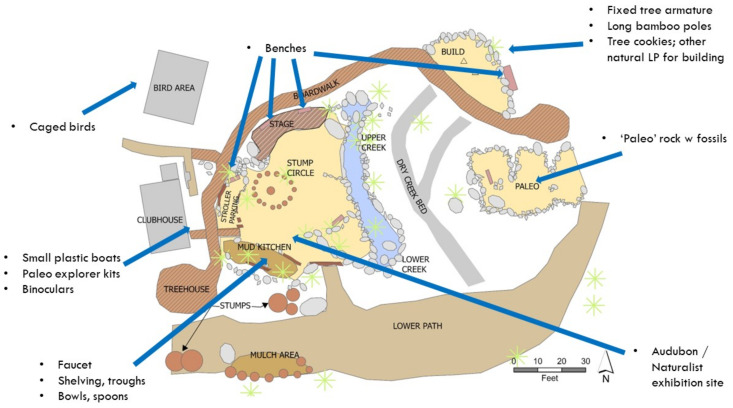
Fixed features and loose parts in *The Backyard*.

**Figure 2 ijerph-19-12661-f002:**
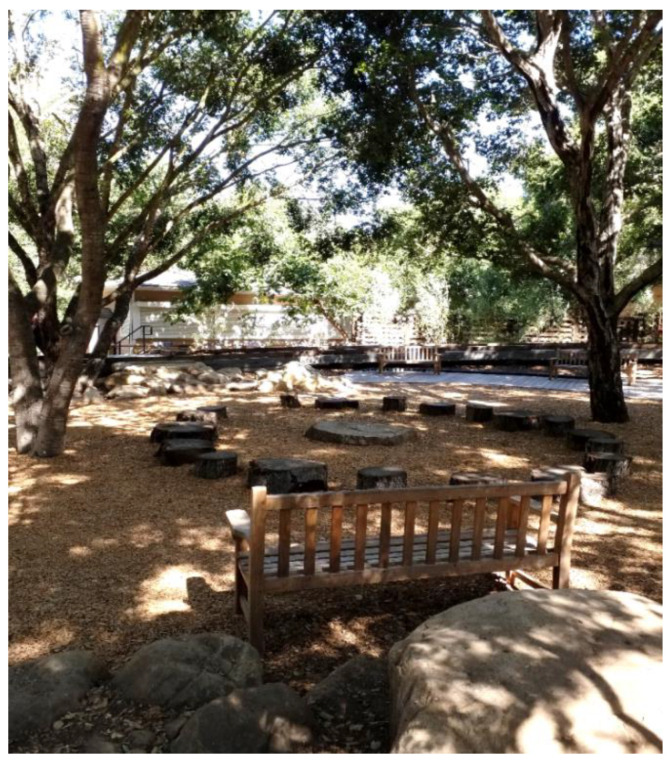
Stump circle and stage areas.

**Figure 3 ijerph-19-12661-f003:**
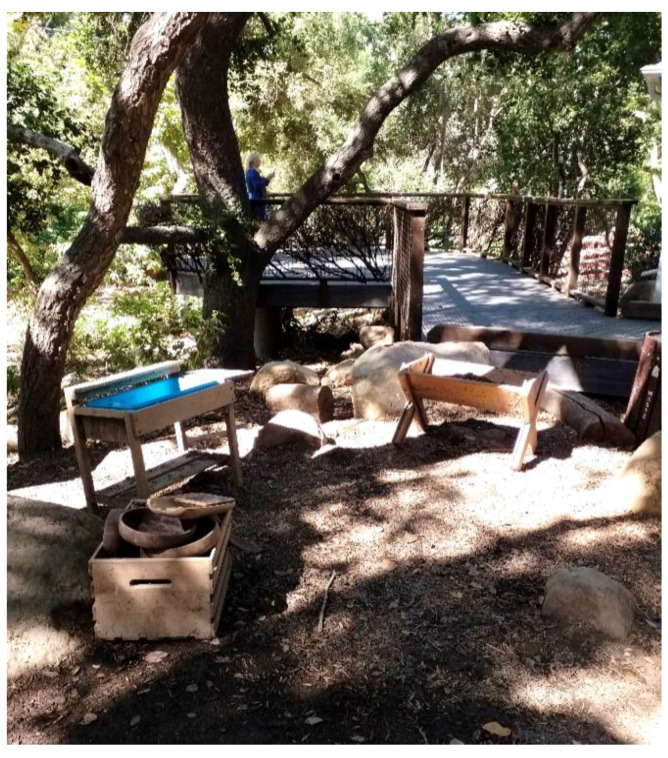
Mud kitchen and treehouse areas.

**Figure 4 ijerph-19-12661-f004:**
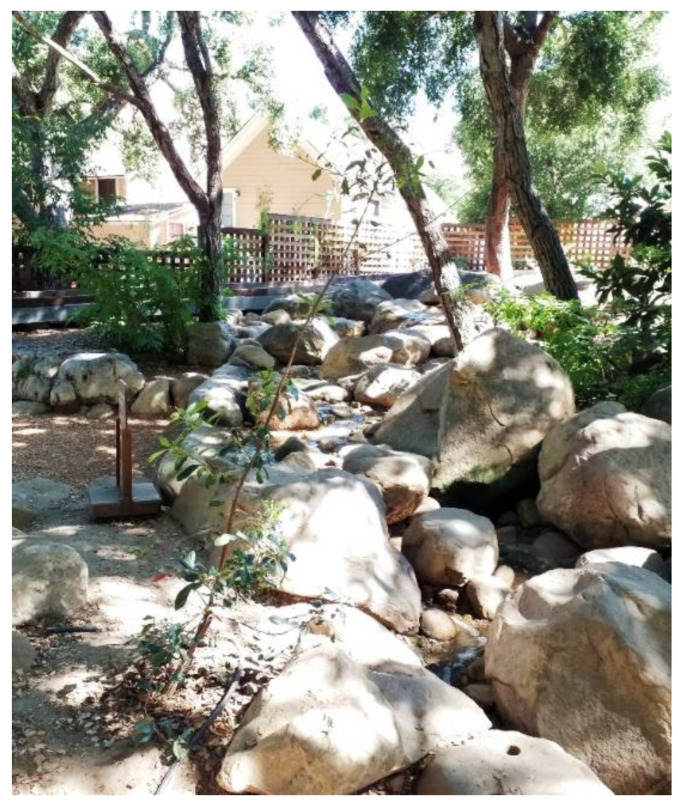
Upper creek area.

**Figure 5 ijerph-19-12661-f005:**
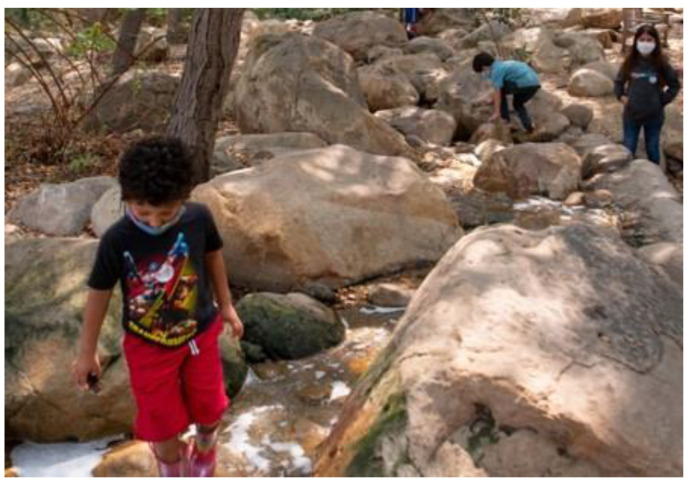
Lower creek area.

**Figure 6 ijerph-19-12661-f006:**
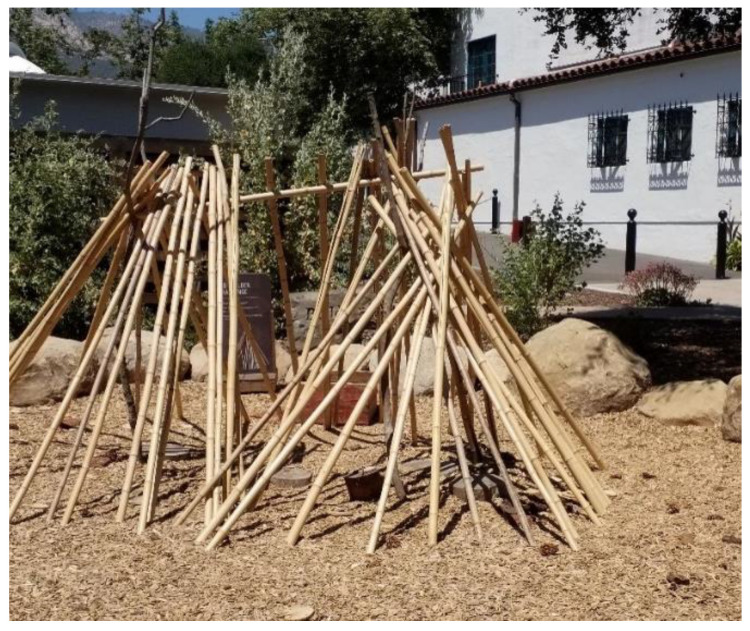
Build area.

**Figure 7 ijerph-19-12661-f007:**
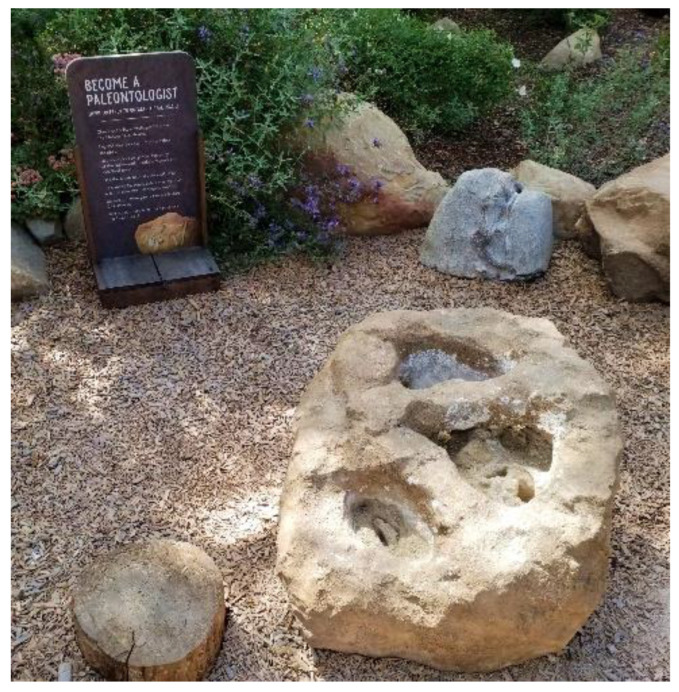
Paleo area.

**Figure 8 ijerph-19-12661-f008:**
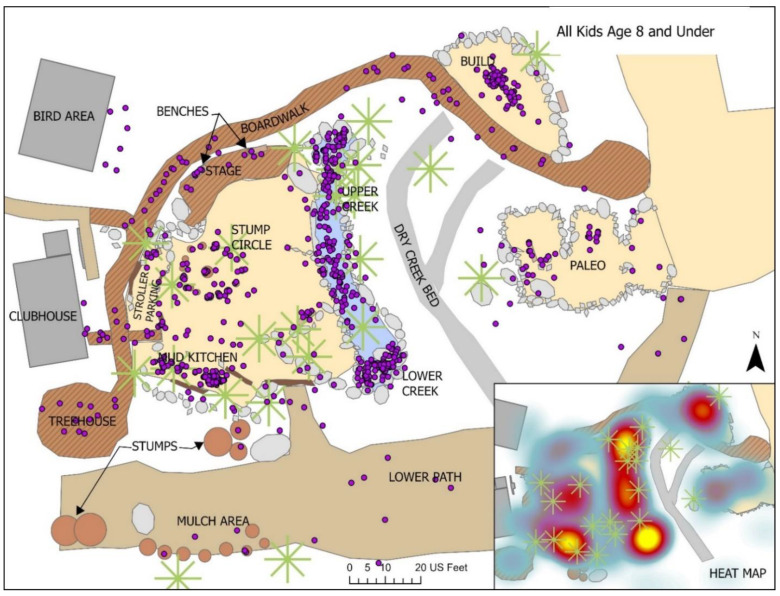
All observed play events for children 8 years and under.

**Figure 9 ijerph-19-12661-f009:**
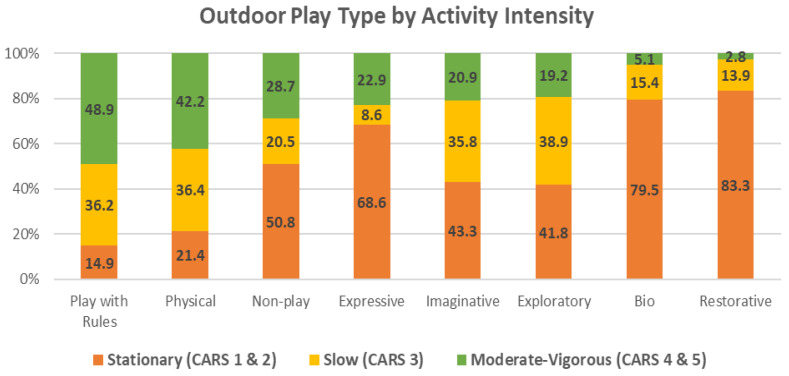
Outdoor play type by activity intensity.

**Figure 10 ijerph-19-12661-f010:**
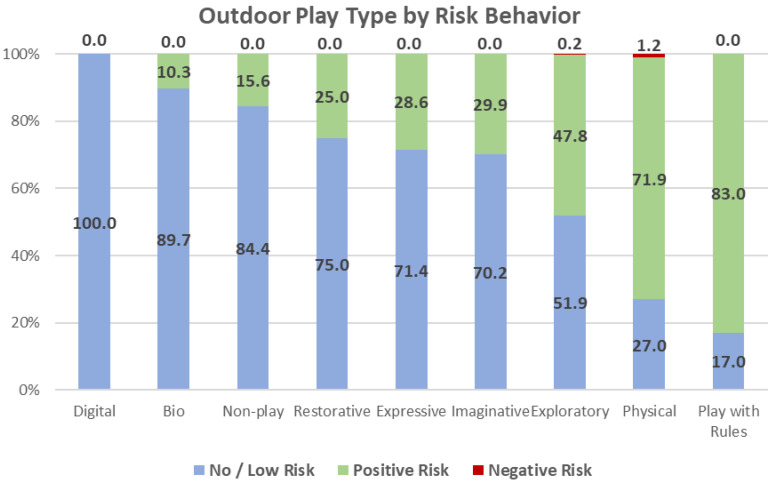
Outdoor play type by risk behavior.

**Figure 11 ijerph-19-12661-f011:**
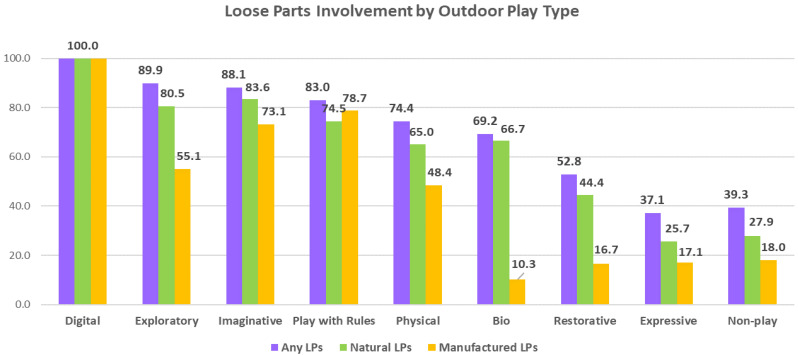
Loose parts’ involvement in outdoor play types.

**Figure 12 ijerph-19-12661-f012:**
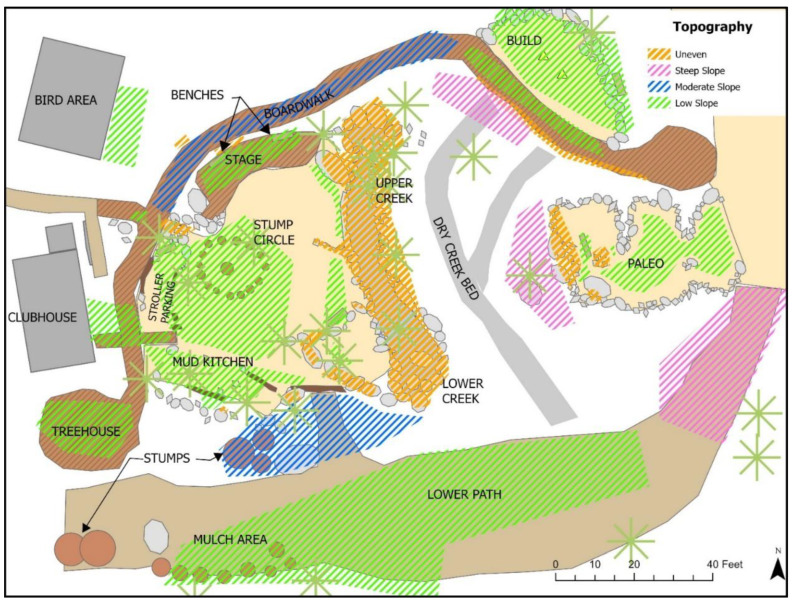
Topography designations for *The Backyard*.

**Figure 13 ijerph-19-12661-f013:**
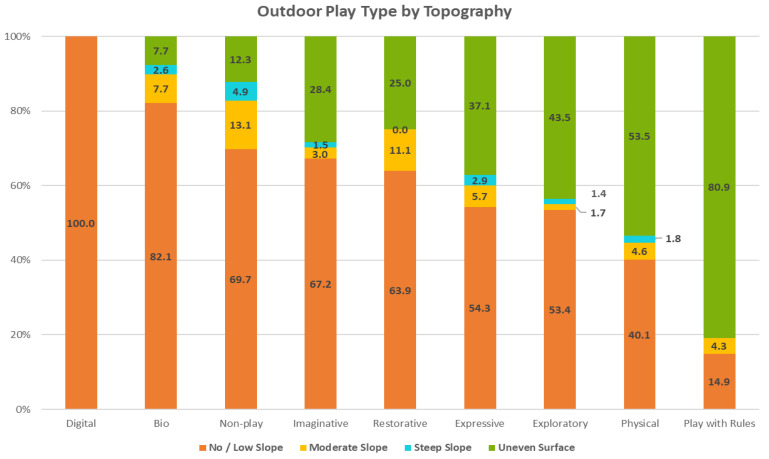
Outdoor play types by topography.

**Figure 14 ijerph-19-12661-f014:**
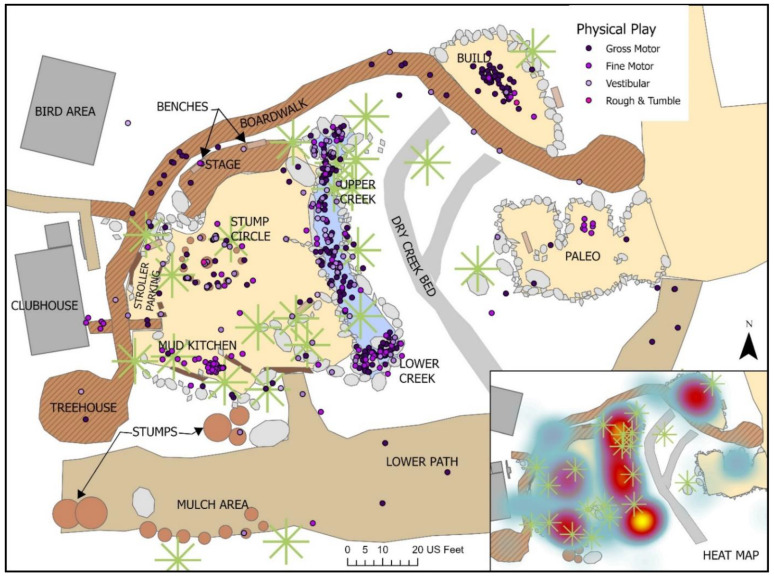
Behavior map of all Physical Play activities.

**Figure 15 ijerph-19-12661-f015:**
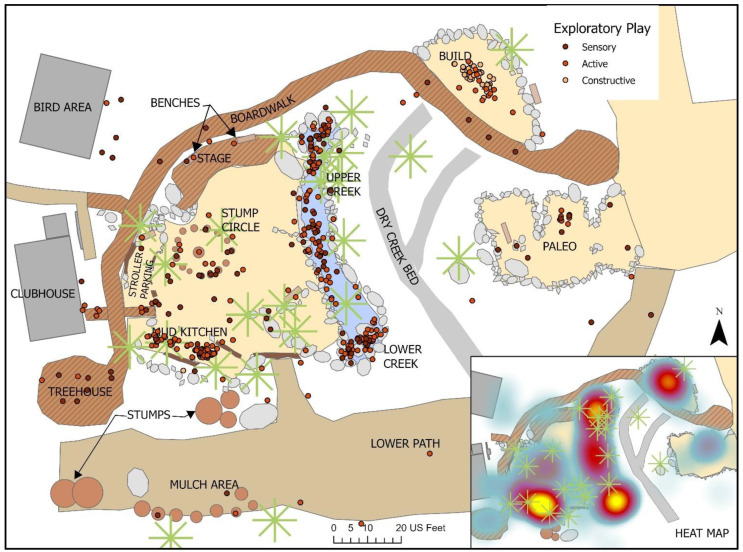
Behavior map of all Exploratory Play activities.

**Figure 16 ijerph-19-12661-f016:**
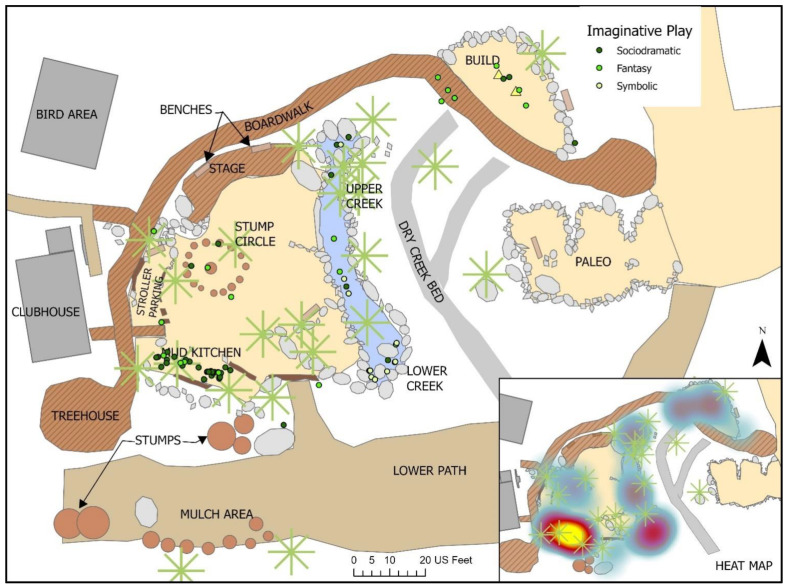
Behavior map of all Imaginative Play activities.

**Figure 17 ijerph-19-12661-f017:**
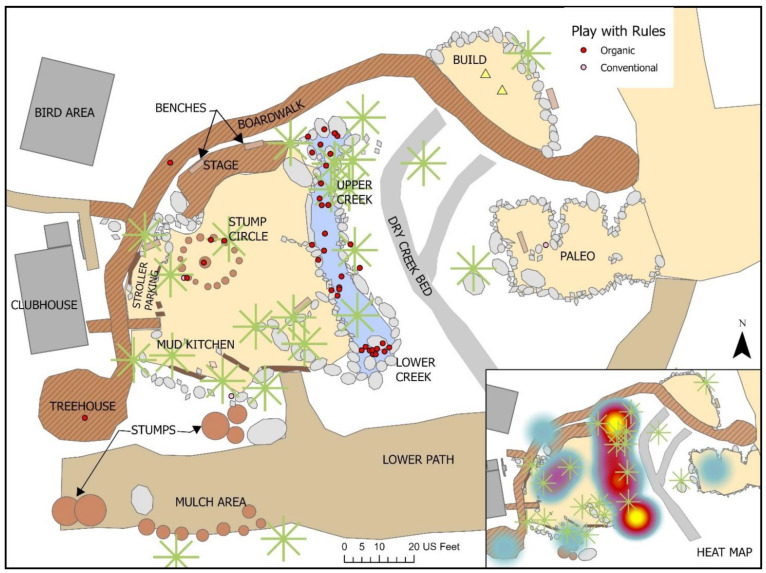
Behavior map of all Play with Rules activities.

**Table 1 ijerph-19-12661-t001:** Descriptives for all observed play events.

Variable	Play Event Frequency (n)	Percentage of Total Play Events (%)
**TOTAL OBSERVED**	**693**	
**GENDER**		
Male	406	58.6
Female	287	41.4
**AGE**		
3–8 y	607	87.6
0–2 y	86	12.4
**PLAY TYPES**		
Physical	434	62.6
Exploratory	416	60.0
Imaginative	67	9.7
Play with Rules	47	6.8
Bioplay	39	5.6
Expressive	35	5.1
Restorative	36	5.2
Digital	1	0.1
Non-play	122	17.6
**PHYSICAL ACTIVITY INTENSITY (CARS)**		
1 Stationary	76	11.0
2 Stationary w Limb Movement	176	25.4
3 Slow	231	33.3
4 Moderate	171	24.7
5 Fast/Quick	39	5.6
**PHYSICAL ACTIVITY INTENSITY (CARS Condensed)**		
Stationary (1 and 2)	252	36.4
Slow (3)	231	33.3
Mod-Quick (4 and 5)	210	30.3
**RISK BEHAVIOR**		
Non/Very Low	322	46.5
Low Positive	239	34.5
Moderate Positive	122	17.6
High Positive	4	0.6
Low Negative	1	0.1
Moderate Negative	3	0.4
High Negative	1	0.1
Risk Appraisal	1	0.1
**RISK BEHAVIOR (Condensed)**		
No/Low Risk	322	46.5
Positive Risk	366	52.8
Negative Risk	5	0.7
**INVOLVED NATURAL ELEMENTS**		
Yes	579	83.6
No	114	16.5
**INVOLVED MANUFACTURED ELEMENTS**		
Yes	514	74.2
No	179	25.8
**INVOLVED LOOSE PARTS**		
Yes	498	71.9
No	195	28.1
**INVOLVED NATURAL LOOSE PARTS**		
Yes	433	62.5
No	260	37.5
**INVOLVED MANUFACTURED LOOSE PARTS**		
Yes	312	45.0
No	381	55.0
**TOPOGRAPHY**		
No/Low Slope	346	49.9
Moderate Slope	32	4.6
Steep Slope	14	2.0
Uneven Surface	301	43.4

**Table 2 ijerph-19-12661-t002:** Outdoor play type activity details and prevalent locations.

OUTDOOR PLAY TYPES				PLAY EVENT DETAILS			MOST PREVALENT SETTINGS
	# Play Events where OPT Involved	% of Primary OPT ^1^	% Involved of All Play Events ^1^	Paired Most Often with:			
**Physical**	434		62.6%	Exploratory (60.2%)	Physical (20.3%)	Play with Rules (7.5%)	
				** *Typical Activities:* **			
*gross motor*	305	70.3%	44.0%	climbing up and down boulders; building structures; running down ramps			creek, build area, slopes
*fine motor*	118	27.2%	17.0%	manipulating boats in creek; play with water/mud in kitchen			creek, mud kitchen
*vestibular*	88	20.3%	12.7%	balancing on boulders, logs and stumps			creek, stump circle
*rough and tumble*	2	0.5%	0.3%	play fighting			n/a
**Exploratory**	416		60.0%	Physical (62.4%)	Imaginative (14.1%)	Bio Play (8.0%)	
				** *Typical Activities:* **			
*sensory*	108	26.0%	15.6%	exploring properties of water; looking for/examining LP, watching wildlife			creek, diverse locations
*active*	275	66.1%	39.7%	experimenting with boats in water; filling/stirring bowls of mud/water/NLP			creek, mud kitchen
*constructive*	43	10.3%	6.2%	building bamboo structures; building water dams in creek			build area, creek
**Imaginative**	67		9.7%	Exploratory (82.8%)	Physical (15.6%)		
				** *Typical Activities:* **			
*symbolic*	9	13.4%	1.3%	playing pretend with various LPs			diverse locations
*sociodramatic*	38	56.7%	5.5%	cooking and ‘house’ or ‘restaurant’ play; pretending to pilot boats			mud kitchen, creek
*fantasy*	20	29.9%	2.9%	playing superheroes, wizards, swords, animals, royalty, monsters			diverse locations
**Play with Rules**	47		6.8%	Physical (69.0%)	Exploratory (28.6%)		
				** *Typical Activities:* **			
*conventional*	4	8.5%	0.6%	Hide n Seek; Duck, Duck, Goose			stump circle; diverse locations
*organic*	43	91.5%	6.2%	racing boats down creek; made up games; obstacle course			creek; open spaces
**Bio**	39		5.6%	Exploratory (85.7%)	Non-Play (5.7%)		
				** *Typical Activities:* **			
*plants*	4	10.3%	0.6%	digging for/collecting acorns; feeling plant leaves; smelling flowers			diverse locations
*wildlife*	35	89.7%	5.1%	observing/interacting with exhibit birds and reptiles; following butterflies			exhibit area; bird cages; diverse locations
*care*	0	0.0%	0.0%	n/a			n/a
**Expressive**	35		5.1%	Non-Play (56.7%)	Physical (23.3%)	Exploratory (10.0%)	
				** *Typical Activities:* **			
*performance*	3	8.6%	0.4%	singing/dancing for or with others			stage, boardwalk, creek
*artistic*	0	0.0%	0.0%	n/a			n/a
*language*	6	17.1%	0.9%	singing/talking to themselves; making up/telling stories; telling jokes			creek, boulder area
*conversation*	27	77.1%	3.9%	social/casual discussions with peers, adults, staff			paleo area benches; diverse locations
**Restorative**	36		5.2%	Exploratory (43.3%)	Non-Play (23.3%)	Physical (16.7%)	
				** *Typical Activities:* **			
*resting*	17	47.2%	2.5%	sitting on boulders, stumps, benches; lying in shade, under trees/structures			creek, stump circle, diverse locations
*retreat*	0	0.0%	0.0%	n/a			n/a
*reading*	1	2.8%	0.1%	reading books brought with them			n/a
*onlooking*	21	58.3%	3.0%	watching other kids, adults from the edges, from a distance			boundaries; behind trees/bushes; from seats
**Digital**	1		0.1%	Physical (100%)			
				** *Typical Activities:* **			
*device*	1	100.0%	0.1%	playing with cell phone			seating area on stage
*augmented*	0	0.0%	0.0%	n/a			n/a
*embedded*	0	0.0%	0.0%	n/a			n/a
**Non-play**	122		17.6%	Physical (31.5%)	Exploratory (30.1%)	Expressive (23.3%)	
				** *Typical Activities:* **			
*self care*	19	15.6%	2.7%	Putting on/tying up shoes; changing out of wet clothes; cleaning face/hands			diverse locations; often on/by seating
*nutrition*	42	34.4%	6.1%	eating lunch/snacks; drinking beverages			diverse locations; often on/by seating
*distress*	2	1.6%	0.3%	crying, arguing with/resisting adult; small injuries			diverse locations
*aggression*	0	0.0%	0.0%	n/a			n/a
*transition*	56	45.9%	8.1%	walking between play settings; entering/exiting; looking for place to sit			paths, open spaces b/w settings, exit areas
*other*	5	4.1%	0.7%	negotiating with parents; collecting items to leave; returning LPs			diverse locations

^1^ Note: Because a single play event could be coded with two OPT subtypes, percentages across subtypes in a single Primary OPT can sum to more than 100%. OPT = Outdoor Play Type; NLP = Natural Loose Parts; LP = Loose Parts.

**Table 3 ijerph-19-12661-t003:** Associations with outdoor play types for select environmental variables using Pearson, Chi^2^ and Fisher’s exact tests.

PHYSICAL PLAY	Loose Parts		Loose Natural		Loose Mfgd		Topography				*Totals*
	*N*	*Y*	*N*	*Y*	*N*	*Y*	*No/Low Slope*	*Mod Slope*	*Steep Slope*	*Uneven*	
No Count	84	175	108	151	157	102	172	12	6	69	259
*Expected Count*	*72.9*	*186.1*	*97.2*	*161.8*	*142.4*	*116.6*	*129.3*	*12*	*5.2*	*112.5*	
Yes Count	111	323	152	282	224	210	174	20	8	232	434
*Expected Count*	*122.1*	*311.9*	*162.8*	*271.2*	*238.6*	*195.4*	*216.7*	*20*	*8.8*	*188.5*	
**Total**	**195**	**498**	**260**	**433**	**381**	**312**	**346**	**32**	**14**	**301**	**693**
Pearson chi^2^ (p)	3.7709	0.052 +	3.0837	0.079	5.3137	**0.021 ***	49.5331	**0.000 *****			
Cramer’s V	0.0738		0.0667		0.0876		0.2674				
Fisher’s exact		0.055 +		0.089		**0.022 ***		**0.000 *****			
**EXPLORATORY PLAY**	**Loose Parts**		**Loose Natural**		**Loose Mfgd**		**Topography**				** *Totals* **
	** *N* **	** *Y* **	** *N* **	** *Y* **	** *N* **	** *Y* **	** *No/Low Slope* **	** *Mod Slope* **	** *Steep Slope* **	** *Uneven* **	
No	153	124	179	98	194	83	124	25	8	120	277
*Expected*	*77.9*	*199.1*	*103.9*	*173.1*	*152.3*	*124.7*	*138.3*	*12.8*	*5.6*	*120.3*	
Yes	42	374	81	335	187	229	222	7	6	181	416
*Expected*	*117.1*	*298.9*	*156.1*	*259.9*	*288.7*	*187.3*	*207.7*	*19.2*	*8.4*	*180.7*	
**Total**	**195**	**498**	**260**	**433**	**381**	**312**	**346**	**32**	**14**	**301**	**693**
Pearson chi^2^ (p)	167.547	**0.000 *****	144.5961	**0.000 *****	42.2694	**0.000 *****	23.5993	**0.000 *****			
Cramer’s V	0.4917		0.4568		0.247		0.1845				
Fisher’s exact		**0.000 *****		**0.000 *****		**0.000 *****		**0.000 *****			
**IMAGINATIVE PLAY**	**Loose Parts**		**Loose Natural**		**Loose Mfgd**		**Topography**				** *Totals* **
	** *N* **	** *Y* **	** *N* **	** *Y* **	** *N* **	** *Y* **	** *No/Low Slope* **	** *Mod Slope* **	** *Steep Slope* **	** *Uneven* **	
No	187	439	249	377	363	263	301	30	13	282	626
*Expected*	*176.1*	*449.9*	*234.9*	*391.1*	*344.2*	*281.8*	*312.5*	*28.9*	*12.6*	*271.9*	
Yes	8	59	11	56	18	49	45	2	1	19	67
*Expected*	*18.9*	*48.1*	*25.1*	*41.9*	*36.8*	*30.2*	*33.5*	*3.1*	*1.4*	*29.1*	
**Total**	**195**	**498**	**260**	**433**	**381**	**312**	**346**	**32**	**14**	**301**	**693**
Pearson chi^2^ (p)	9.6243	**0.002 ****	14.0867	**0.000 *****	23.6824	**0.000 *****	8.8251	**0.032 ***			
Cramer’s V	0.1178		0.1426		0.1849		0.1128				
Fisher’s exact		**0.001 ****		**0.000 *****		**0.000 *****		**0.029 ***			
**PLAY WITH RULES**	**Loose Parts**		**Loose Natural**		**Loose Mfgd**		**Topography**				** *Totals* **
	** *N* **	** *Y* **	** *N* **	** *Y* **	** *N* **	** *Y* **	** *No/Low Slope* **	** *Mod Slope* **	** *Steep Slope* **	** *Uneven* **	
No	187	459	248	398	371	275	339	30	14	263	646
*Expected*	*181.8*	*464.2*	*242.4*	*403.6*	*355.2*	*290.8*	*322.5*	*29.8*	*13.1*	*280.6*	
Yes	8	39	12	35	10	37	7	2	0	38	47
*Expected*	*13.2*	*33.8*	*17.6*	*29.4*	*25.8*	*21.2*	*23.5*	*2.2*	*0.9*	*20.4*	
**Total**	**195**	**498**	**260**	**433**	**381**	**312**	**346**	**32**	**14**	**301**	**693**
Pearson chi^2^ (p)	3.0817	0.079	3.09	0.079	23.1361	**0.000 *****	29.6794	**0.000 *****			
Cramer’s V	0.0667		0.0668		0.1827		0.2069				
Fisher’s exact		0.093		0.087		**0.000 *****		**0.000 *****			
**BIO PLAY**	**Loose parts**		**Loose Natural**		**Loose Mfgd**		**Topography**				** *Totals* **
	** *N* **	** *Y* **	** *N* **	** *Y* **	** *N* **	** *Y* **	** *No/Low Slope* **	** *Mod Slope* **	** *Steep Slope* **	** *Uneven* **	
No	183	471	247	407	346	308	314	29	13	298	654
*Expected*	*184*	*470*	*245.4*	*408.6*	*359.6*	*294.4*	*326.5*	*30.2*	*13.2*	*284.1*	
Yes	12	27	13	26	35	4	32	3	1	3	39
*Expected*	*11*	*28*	*14.6*	*24.4*	*21.4*	*17.6*	*19.5*	*1.8*	*0.8*	*16.9*	
**Total**	**195**	**498**	**260**	**433**	**381**	**312**	**346**	**32**	**14**	**301**	**693**
Pearson chi^2^ (p)	0.1414	0.707	0.3087	0.578	20.1789	**0.000 *****	21.6026	**0.000 *****			
Cramer’s V	−0.0143		0.0211		−0.1706		0.1766				
Fisher’s exact		0.715		0.614		**0.000 *****		**0.000 *****			
**EXPRESSIVE PLAY**	**Loose parts**		**Loose Natural**		**Loose Mfgd**		**Topography**				** *Totals* **
	** *N* **	** *Y* **	** *N* **	** *Y* **	** *N* **	** *Y* **	** *No/Low Slope* **	** *Mod Slope* **	** *Steep Slope* **	** *Uneven* **	
No	173	485	234	424	352	306	327	30	13	288	658
*Expected*	*185.2*	*472.8*	*246.9*	*411.1*	*361.8*	*296.2*	*328.5*	*30.4*	*13.3*	*285.8*	
Yes	22	13	26	9	29	6	19	2	1	13	35
*Expected*	*9.8*	*25.2*	*13.1*	*21.9*	*19.2*	*15.8*	*17.5*	*1.6*	*0.7*	*15.2*	
**Total**	**195**	**498**	**260**	**433**	**381**	**312**	**346**	**32**	**14**	**301**	**693**
Pearson chi^2^ (p)	21.9737	**0.000 *****	21.2575	**0.000 *****	11.5747	**0.001 ****	0.7	0.873			
Cramer’s V	−0.1781		−0.1751		−0.1292		0.0318				
Fisher’s exact		**0.000 *****		**0.000 *****		**0.001 ****		0.655			
**RESTORATIVE PLAY**	**Loose parts**		**Loose Natural**		**Loose Mfgd**		**Topography**				** *Totals* **
	** *N* **	** *Y* **	** *N* **	** *Y* **	** *N* **	** *Y* **	** *No/Low Slope* **	** *Mod Slope* **	** *Steep Slope* **	** *Uneven* **	
No	178	479	240	417	351	306	323	28	14	292	657
*Expected*	*184.9*	*472.1*	*246.5*	*410.5*	*361.2*	*295.8*	*328*	*30.3*	*13.3*	*285.4*	
Yes	17	19	20	16	30	6	23	4	0	9	36
*Expected*	*10.1*	*25.9*	*13.5*	*22.5*	*19.8*	*16.2*	*18*	*1.7*	*0.7*	*15.6*	
**Total**	**195**	**498**	**260**	**433**	**381**	**312**	**346**	**32**	**14**	**301**	**693**
Pearson chi^2^ (p)	6.8391	**0.009 ****	5.2702	**0.022 ***	12.3343	**0.000 *****	8.6879	**0.034 ***			
Cramer’s V	−0.0993		−0.0872		−0.1334		0.112				
Fisher’s exact		**0.013 ***		**0.032 ***		**0.000 *****		**0.034 ***			
**DIGITAL PLAY**	**Loose parts**		**Loose Natural**		**Loose Mfgd**		**Topography**				** *Totals* **
	** *N* **	** *Y* **	** *N* **	** *Y* **	** *N* **	** *Y* **	** *No/Low Slope* **	** *Mod Slope* **	** *Steep Slope* **	** *Uneven* **	
No	195	497	259	433	381	311	345	32	14	301	692
*Expected*	*194.7*	*497.3*	*259.6*	*432.4*	*380.5*	*311.5*	*345.5*	*32*	*14*	*300.6*	
Yes	0	1	1	0	0	1	1	0	0	0	1
*Expected*	*0.3*	*0.7*	*0.4*	*0.6*	*0.5*	*0.5*	*0.5*	*0*	*0*	*0.4*	
**Total**	**195**	**498**	**260**	**433**	**381**	**312**	**346**	**32**	**14**	**301**	**693**
Pearson chi^2^ (p)	0.3921	0.531	1.6678	0.197	1.2229	0.269	1.0043	0.800			
Cramer’s V	0.0238		−0.0491		0.042		0.0381				
Fisher’s exact		1.000		0.375		0.450		1.000			
**NON PLAY**	**Loose parts**		**Loose Natural**		**Loose Mfgd**		**Topography**				** *Totals* **
	** *N* **	** *Y* **	** *N* **	** *Y* **	** *N* **	** *Y* **	** *No/Low Slope* **	** *Mod Slope* **	** *Steep Slope* **	** *Uneven* **	
No	121	450	172	399	281	290	261	16	8	286	571
*Expected*	*160.7*	*410.3*	*214.2*	*356.8*	*313.9*	*257.1*	*285.1*	*26.4*	*11.5*	*248*	
Yes	74	48	88	34	100	22	85	16	6	15	122
*Expected*	*34.3*	*87.7*	*45.8*	*76.2*	*67.1*	*54.9*	*60.9*	*5.6*	*2.5*	*53*	
**Total**	**195**	**498**	**260**	**433**	**381**	**312**	**346**	**32**	**14**	**301**	**693**
Pearson chi^2^ (p)	77.4259	**0.000 *****	75.6734	**0.000 *****	43.5725	**0.000 *****	73.9229	**0.000 *****			
Cramer’s V	−0.3343		−0.3304		−0.2507		0.3266				
Fisher’s exact		**0.000 *****		**0.000 *****		**0.000 *****		**0.000 *****			

+ *p* ~ 0.05, * *p* < 0.05, ** *p* < 0.01, *** *p* < 0.001.

## Data Availability

Data supporting reported results will be available through the Cornell Institute for Social and Economic Research www.archive.ciser.cornell.edu, accessed on 1 July 2022.

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
