# Peer review of "Playing in ‘The Backyard’: Environmental Features and Conditions of a Natural Playspace Which Support Diverse Outdoor Play Activities among Younger Children"

_ijerph, 2022, doi:10.3390/ijerph191912661_

Round 1

Reviewer 1 Report

This is an interesting manuscript about healthy development among young children. The initial part (1. Introduction) is well argued thanks also to a pertinent, updated and useful bibliography. The methodology used is effective. Charts, maps and tables are well crafted and useful. Reflections on variations in topography and their effects are very interesting and valid from a scientific point of view.

To improve the work, however, I recommend implementing some considerations: we are discussing the design of new spaces dedicated to children, but how can we improve existing spaces without the required environmental characteristics? The transformation of what already exists is very important. In this sense, it might be useful to briefly propose a successful case study or write some considerations. The proposed case study (Backyard at the Santa 183 Barbara Museum of Natural History) is interesting but perhaps the scope is too narrow in order to draw meaningful deductions.

Data were collected during a week in July 2019. It seems too short a time. Why not make a comparison with the data collected at a different time of the year?

Authors still need to reread carefully the text. There are several typos, such as "typography" instead of "topography" (line 331). See also lines 894, 963, 974. Compliance with editorial rules must always be consistent (see line 1017). The indication of [46] associated with the quote from Simon Nicholson is missing (see line 90).

Reviewer 2 Report

-The abstract should be a total of 200 words maximum, your abstract has 280 words.

- Keyword: try to avoid redundances with the word "play".

- Please check: In the text, reference numbers should be placed in square brackets [ ], and placed before the punctuation; for example [1], [1–3] or [1,3]. In the whole text

- Check doble space for example line 62, 96, 102,159, 288, 342, 418, 464,510, 637, 645, 652, 708, 721, 729, 742, 748, 749, 813.

- Check the initial page, line 136 for example [5] (p. 10). or [6] (pp. 101–105).

- Line 163 missed punctuation

-line 234 citation just one surname no two

- line 272, 274, etc., etc., no necessary mention (See….), when referring to a figure, table, or graph.

- space line 272

- figure X ??? line 414

- line 477 uses one digit, not two

- In the Discussion, there are only two authors. It is important to substantiate the findings found with what the literature or other investigations provide us, for which the Discussion section must be formulated again.

- In the References: there are two numerations. Also, Author: The name is written inverted (surname and the initials of the name, followed by a period). When there is more than one author, they are separated by a semicolon, for example, Cotton, F. A. or Chandler, J. P.: Levine, S. M.

Reviewer 3 Report

Review Article: Playing in «The Backyard”: Environmental features and conditions of a Natural playscape which support diverse outdoor play activities among younger children

Brief summary:

The article addresses important fields of health promoting playscapes for children emphasizing the value of nature-based environments. Generally, the article has a sound approach to this specific field of research, is well written and applies appropriate research methodology, but anyhow shows some weaknesses. The novelty of the study could be questioned as the aims of the study are rather diffuse and the focus of the study is more related to the applied methods. The introduction and background of the study is well documented by appropriate references focusing on different aspects and practice of play categories. This seems to be is somehow too comprehensive in a text covering almost three pages. These play categories seem to be coinciding with the methodology for observing outdoor play types and could constitute the base for the observation of play forms.  More seriously: the knowledge gap is not presented and discussed – neither are the aims of the study that will focus on filling the knowledge gap. This should be better reflected, presented, and discussed. Overall, the focus is mainly on methods and the “why” approach should be more clearly both in abstract and introduction. The selected methods for the study seems appropriate, but should more clearly be related to project aims and research questions.  Behaviour maps of different play categories are very welcome! – but the relation to nature characteristics should be clearer and more detailed – only topography and general designations are indicated. What about vegetation – trees and different vegetation, animals, etc. that afforded the different play activities?  Results presented in tables should be commented in text– highlight the main findings. Discussion is too long with only one reference. Here the main findings should be discussed related to the aims of the study and previous studies. Strength and weaknesses of the study are not discussed. What about ethical guidelines for studying children? Although this study has some weaknesses, it should be encouraged to be processed into a better and accepted version.

General comments
The topic of the article is sound and appropriate and focuses on important perspectives for children’s playscapes, especially highlighting natural playscapes. However, the study has some areas of weakness, especially in presenting main aims and research questions (RQ) for the study – which should be based on an identified gap in knowledge. The selected methods should more clearly address the RQ’s and the study design better and fitting the target groups: the children in different ages. The target groups were categorized as children aged 0-17 years and split in smaller age groups. However, - the results were limited to the age group 3-8 years, - sometimes 0-2 years, which seems to be a rather inhomogeneous age group? Criteria for differentiating between children are diffuse – and who is doing what? Mapping tools seem to be appropriate but maps showing nature, like biotopes are missing – only topography is included – but lacking the exact information of height curves and equidistance. Photos of nature playscape and landscape forms would help. Methodology for observing outdoor play (TOPO) seems relevant. Results: Too much information in figures and tables. Suggestion: base your results on the figures and add information from the tables in text only. Figures with bars and maps give good information. The tables are too much of less important information. Discussion: To much of telling and explaining. Shorten and stick to main aims and RQ’s and discuss how your results bring in new knowledge to the research field – referring to literature.

Add a paragraph on ethics, especially national and international guidelines for research with children.

Add a paragraph on strength and weakness of the study.

Conclusion: Add an assessment of how your study has contributed to new knowledge to the research field.

Search for more recent studies on Nature-based playscapes and children’s physical and mental development and health

Specific comments:

Page 6-25: Results: Consider dropping the tables and include the results shortened in text

Focus the results for answering the main aim and research questions.
